# Estimating Total Energy Expenditure for Fire-Fighters during Large Scale Disaster Response Training Using a Tri-Axial Accelerometer

**DOI:** 10.3390/nu13082789

**Published:** 2021-08-14

**Authors:** Nao Koizumi, Yutaro Negishi, Hitomi Ogata, Randeep Rakwal, Naomi Omi

**Affiliations:** 1Graduate School of Comprehensive Human Sciences, University of Tsukuba, 1-1-1 Tennodai, Tsukuba 305-8574, Japan; s1830459@s.tsukuba.ac.jp (N.K.); negiro424@gmail.com (Y.N.); 2Graduate School of Humanities and Social Sciences, Hiroshima University, Higashi-Hiroshima, Hiroshima 739-8521, Japan; hogata@hiroshima-u.ac.jp; 3Faculty of Health and Sport Sciences, University of Tsukuba, 1-1-1 Tennodai, Tsukuba 305-8574, Japan; plantproteomics@gmail.com

**Keywords:** energy expenditure, tri-axis accelerometer, high-level physical activity, energy required for firefighters

## Abstract

The present study was conducted to estimate total energy expenditure (TEE) of fire-fighters using tri axial-accelerometers in conjunction with an activity log survey on a large number of subjects undergoing training mimicking a large-scale disaster. Subjects were 240 fire-fighters participating in a two-day fire-fighting training dedicated to large-scale natural disasters. Data was analyzed by job type of activity group and the job rank, and by comparing the average. The average TEE of the total survey training period is about 3619 (±499) kcal, which is the same value of expenditure for professional athletes during the soccer game season. From the activity group, the rescue and other teams consumed significantly more energy than the fire and Emergency Medical Team (EMS) teams. From the job rank, Fire Captain (conducting position) consumed significantly lower energy than the Fire Lieutenant and Fire Sergeant. Furthermore, it was found that a middle position rank consumed the most energy. This research supports a need to reconsider the current rescue food (and protocols) to supplement the energy expenditure of fire-fighters. In addition, since there was a significant difference between the job type and the job rank, it is necessary to examine the energy amount and shape suitable for each.

## 1. Introduction

When a large disaster occurs, fire-fighters are expected to engage in disaster rescue activities of high intensity and for a long period under harsh environmental conditions. The first 48 h after the disaster are especially critical, as the priority is on the saving victims’ lives and this is the time period where the fire-fighters usually work without sleep and rest. In Japan, when a large-scale disaster occurs, emergency fire response teams are organized by the fire department and fire-fighters are dispatched nationwide for various disaster-relief activities; this implies working away from the home base and the necessity of transporting all necessary fire-fighting equipment and resources in order to carry out multiple duties longer than usual duty hours.

A critical requirement in disaster-relief operations comes down to a very basic requirement, namely food or ‘nutrition’ for the fire-fighters. “Food” is especially required to maintain the body condition of fire-fighters in such emergency situations, and supporting them for their enhanced role as rescue personnel in the disaster zone. Under such high-intensity work/abnormal situations it is understandable that there might be a less than sufficient food supply and this lack of food (energy) creates an adverse effect on the fire-fighters. A recent study has looked into the indices of metabolic and cardiovascular health in wildland fire-fighters, pre- and post-season maladaptive changes in adipose tissue, blood lipids and hepatic function [1]. Akano et al. (2013) reported that about half of fire-fighting personnel showed ‘poor physical condition’ during the Great East Japan earthquake in 2011 [2]. A report from the Fire and Disaster Management Agency issued in 2012 revealed that it is the responsibility of each fire department to make food, water, personal equipment, etc., available for more than 72 h [3]. However, complying with this condition is difficult, and it was found that the stockpile of each fire department headquarters is not sufficient [4]. The same research also noted that the fire department needed other technical information such as the energy requirements of fire-fighters, which has a role in preparing the appropriate food [4]. With regards to the energy intake, it was reported that the fire-fighters consumed an inadequate amount of total calories, including some nutrients, compared to a military dietary reference intake [5]. These studies also led us to question and find out how much energy is consumed in the actual activity for the preparation during an actual disaster. The firefighters have different roles (activities) depending on their positions, and it was hypothesized that there would be differences in energy consumption. Moreover, based on the division of roles and the command system [6], it is our belief that there needs to be a comparison by job type of activity group and the job rank.

Japan is one of the countries most affected by natural disasters such as typhoons, heavy rain, floods, landslides, earthquakes, tsunami, snow and volcanic eruptions, etc. [7]. Large-scale disasters in Japan usually result in multiple damages across both rural and urban areas, ranging from mountainous to coastal areas. Therefore, large-scale disasters cause various accidents for example huge fires, building collapses, flood damages, landslides, etc. Japanese fire departments provide a variety of services, such as fire-fighting, emergency medical service (EMS), rescue service and backup units in the event of a disaster. A National Fire Service Team for Disaster Response (NFSTDRT) is usually created by the Japanese government immediately following a large disaster. The NFSTDRT was founded in 1995, after the Great Hanshin-Awaji Earthquake and institutionalized by the Fire Defense Organization Law, as amended in 2003. Officially, NFSTDRT was founded in 2004 [3]. Since 2004, the Commissioner of the Fire and Disaster Management Agency has the authority to mobilize and control fire-fighting teams in the event of a large-scale disaster or accident [3]. In the case of the Great East Japan Earthquake, for the first 88 days of 11 March 2011 to 6 June 2011, the emergency fire-fighting support team constituted of a total of 30,684 personnel and carried out numerous disaster response activities. In addition, from the lessons learned from the Great East Japan Earthquake [8], the Japanese government indicated that the number of emergency fire-fighting support team registered targets will be significantly increased to 6000 units in 2018, for the purpose of strengthening the response to the next large disaster [9]. Therefore, more and more emergency fire assistance teams will be registered in the future, and the number of instances where many officers to be dispatched to affected areas as emergency fire responses brigades will increase. As the need for a NFSTDRT is projected to increase in the future, it is expected that it will become increasingly important to understand the members’ activities and manage those activities in light of the available information.

Due to logistic issues and costs, the fire-fighters’ field survey has been limited to-date [10]. For example, the use of the double-labeled water (DLW) test as the gold standard for the measurement of total energy expenditure (TEE) is expensive, and also requires a lot of time other than disaster activities during the study period. Total energy expenditure at the time of disaster activity is presumed to be about 3000 to 4900 kcal from a literature review research [10,11,12,13]. The TEE that has been reported is calculated from a small number of participants, and there is no report on using a large participation group. Omi et al. (2014) [14] estimated that the energy requirement of fire-fighters in the event of a large-scale disaster is about 3000 to 4000 kcal, but it is difficult to determine accurately at present due to a wide range of estimated values. Omi et al. (2014) [14], Parker et al. (2017) [15], and Robertson et al. (2017) [16] tried to verify the physical activity of fire-fighters using new equipment, but the number of samples remained small. Fire-fighters have various roles during a large-scale disaster activity, but there are only a few cases where they are quantitatively verified. In addition, these are the results of data from the fire-fighters only. The data on other personnel such as the paramedics and rescue staff who also work alongside the fire-fighters at disaster sites has not been collected or analyzed, and remains unclear. Furthermore, all ranks cannot take part in these disaster-related activities; moreover, their activity level is different depending on the job position and job rank. Nevertheless, the authors have not been able to find these reports for verification and clarification. Therefore, in this research, the focus was on the measurement of a large number of participants simultaneously, using the accelerometer (AC) method that is actually wearable, and which can divide the time for each activity. Furthermore, we aimed to clarify the characteristics of each type of activity team and by job rank.

Based on the above facts and gaps in research, it is essential for fire-fighters to not only know but also understand their energy expenditure demands during an emergency disaster response management. The aim of this study was therefore to estimate the TEE of fire-fighters by utilizing the accelerometers on a large number of subjects mimicking a large-scale disaster. Therein, the research questions were formulated as: (i) What is the average TEE of fire-fighters during a large-scale disaster activity training? and, (ii) What are the characteristics of the required energy in each job rank and activity group? To fulfil this aim, the scope of study will be delineated to assess the training of large-scale disaster activities. This training is conducted in a masked format that does not reveal the contents to the participants in advance, so it can be said that this training is close to a real disaster. It is anticipated that findings from this research will also provide an academic base to update the rescue food that could support fire-fighters in emergency situations.

## 2. Materials and Methods

### 2.1. Approach

A survey- and questionnaire-based approach was used to estimate the energy expenditure of fire-fighters using the tri-axis accelerometer and activity log records. This study was conducted with the approval of the University of Tsukuba Physical Education Research Ethics Committee (Number: T28-66, Study on the activity and nutrition of Fire service in disaster response, approved 3 October 2016).

### 2.2. Training and Participants 

#### 2.2.1. Survey Target Training

The study targeted fire-fighters comprising a total of 241 research subjects participating in the five 2-day disaster simulation training (i.e., fire-fighters do not know what is in the training protocol). This study carried out five cross-sectional surveys of diverse category research subjects in different situations. The purpose of this research was to calculate a value closer to the actual value by conducting multiple surveys and averaging them. The reason was because each actual disaster has a different weather condition and associated damage. Long-term training was conducted with assumptions that are close to real disasters and whose contents are not informed to participating members in advance. Although it is desirable to measure with an actual disaster, when an actual disaster occurs, there is urgency and it is impossible to attach the devices and perform the measurements. Furthermore, because it is unpredictable, we targeted disaster simulation training which is considered to be the most similar to a real disaster (Appendix A).

#### 2.2.2. Participants

241 officers from 28 fire departments of seven prefectures across Japan were recruited to participate in this study. Males, selected as part of the training members of the National Fire Service Team for Disaster Response, were members who were in good health, passed the physical fitness test at the time of hiring and had completed basic education (excluding those who have been in service for less than half a year). The participants were randomly selected by the head of each institution, which consented to their participation in the survey after the list of participants for each training subjects to the survey was prepared. All participants (n = 241) were male with a mean age of 35 ± 7.3 years, height of 172.5 ± 5.5 cm, weight of 69.7 ± 7.8 kg, and Body Mass Index (BMI) of 23.4 ± 2.1 kg/m^2^. They were organized into rescue, firefighting, EMS and the other teams (the command and back up unit). Each officer is classified by job rank (Fire captain, Fire Lieutenant, Fire Sergeant, Assistant Fire Sergeant and Fire Fighter) (Table 1). Participants were informed and agreed to the details of this study. In addition, a detailed explanation was given to each training director and each participating organization affiliated institution on the purpose and contents of the experiment, and this was started with an agreement.

### 2.3. Procedures and Instrumentation

Demographic data (i.e., body height, body mass, age) for each subject was obtained via a questionnaire. Each subject was required to wear a tri-axial accelerometer (Omron activity meter Active style PRO HJA-750C—23 g, 40 × 52 × 12 mm; OMRON, Kyoto, Japan) as an evaluation method of the AC method during the targeted training. 

### 2.4. Validation of Accelerometer Placement

In previous research, the position of the accelerometer (Active style Pro HJA-750C) is indicated as to be strapped on the waist [17]. Moreover, there were cases where the accelerometer was secured on the waist belt [18] or carried in the chest pocket [10]. The rationale for waist placement is based on the assumption that the movement of the center of gravity of the body is associated with physical energy expenditure. On the other hand, Heil (2002) had also tried to estimate the TEE of a firefighter by attaching it to the chest part [10]. 

In disaster activities like those used for the disaster simulation training targeted in the present study, it is expected that officers will wear fire-resistant clothes on top of standard activity clothes. Furthermore, when searching for refugees in the debris of the disaster site, the officers move in such a way so as to bend their posture to the narrow part. Thus, there is a possibility of breaking the equipment when it is attached on the outside and the possibility of it detaching unintentionally may happen. Therefore, we considered placing the device on the chest area. The base part of the chest pocket (length 16.2 cm, 15.3 cm wide) is located near the waist area (about 10 cm above the belt part) (Figure 1). Additionally, in order to avoid the influence of liquids such as water used by fire extinguishing activity etc., the accelerometer was covered with vinyl material and fixed onto the chest pocket with adhesive tape.

### 2.5. Data Processing Algorithm

Accelerometers (Active style Pro HJA-750C, Omron Healthcare Co., Ltd. Kyoto, Japan) were programmed to save activity data once per 10 sec, which resulted in data files totaling 8640 data points each day. With a measurement frequency of 32 Hz, it is possible to record at epoch length of 10 s value. It records acceleration information on three axes in the longitudinal direction (x axis) and the lateral direction (y axis) vertical direction (z axis) and calculates the resultant acceleration. The dynamic range showing the ratio between the minimum value and the maximum value of identifiable signals is 3 mG to 6 G. It is also possible to capture correct physical activity such as a change in body movement and posture. In addition, it can record high intensity physical activity more accurately than walking [17]. Time interval for recording the acceleration (epoch length) is 60 s, which has been widely used [19], and has also been recommended for a shorter time of recording [20]. 

Devices capable of measuring in a shorter time have been developed, and the validity of the physical activity evaluation of the epoch length value for 10 s has been shown [19]. For that reason, we selected equipment that can measure every 10 s. The data was recorded at each 10 s data point, and it was transformed into METs (i.e., metabolic equivalent) using Omron’s algorism [17,21]. Next, each transformed data point was converted to units of AEE (Activity Energy Expenditure), kcal min/10 s [17,21]. The algorithm embedded in Omron’s equipment is said to display 24 h from 0:00 to 24:00 as the daily energy expenditure. However, in the disaster simulation training, participants performed the rescue activity from 11:00 to 11:00 the next day, so we calculated from 11:00 to 11:00 as one day.

### 2.6. Activity Log

The activity log of the research was recorded by filling the survey form distributed to the research participants. They recorded their activity log every 30 min. The participants were asked to describe their activities in the activity log. Because the Tri-axial Accelerometer only shows the intensity, we asked each participant to write down what kind of activity they actually did.

### 2.7. Analysis

All data were analyzed using windows software SPSS (version 24). Descriptive statistics (mean ± standard deviation (SD)) were initially calculated for all measured variables. One-way analysis of variance was used for mean-to-mean significance testing, and Tukey’s HSD method was used for multiple comparisons.

## 3. Results

### 3.1. Participants, Location and Weather Condition

A total of 241 participants joined the study; however, one participant withdrew from the study due to technical issues with the accelerometer, and thus data from 240 participants was collected and analyzed in this study (n = 240) (Table 1). All trainings were held 24 h a day from the morning to the next morning in the Kanto region of Japan. Rainfall affected their activity during training A, B and C. In the A session, some training was interrupted due to rainfall (Appendix A).

### 3.2. Total Energy Expenditure (TEE)

The average of the total survey training TEE is about 3619 (±499) kcal. A comparison with activity group is shown in Figure 2. The TEE of the rescue team had an average of 3707 (±491) kcal, the average of the fire-fighting team was 3290 (±342) kcal, the EMS team had an average of 3308 (±318) kcal, and the other teams had 3774 (±575) kcal. The average energy expenditure was the highest for the other teams and the lowest for fire-fighting team. The TEE of the rescue team and the other teams were significantly higher (*p* < 0.05) than the fire-fighting team and EMS team. 

Comparison by job rank is shown in Figure 3. TEE of the Fire Captain is 3329 (±272) kcal on average, 3692 (±573) kcal on average for the Fire Lieutenant, 3691 (±482) kcal on average for the Fire Sergeant, and 3489 (±386) kcal on average for the Assistant Fire Sergeant. The average was 3546 (±378) kcal for the Fire-Fighter. The average energy expenditure was the highest for the Fire Lieutenant and lowest for the Fire Captain. TEE of the Fire Captain was significantly lower (*p* < 0.05) than the Fire Lieutenant and Fire Sergeant. 

### 3.3. Intensity of Activity 

The intensity of activity was classified into SED, LPA and MPA+VPA by each activity group. Similar to previous studies, the intensity of activity was determined by activity count cut points for sedentary (SED; >1.49 METs), light (LPA; >1.5–2.99 METs), moderate (MPA; >3.0–5.99 METs) and vigorous-intensity physical activity (VPA; >6 METs) [22]. The mean percent of time spent in each intensity category included: sedentary (39%), light (44.5%), and moderate/vigorous (16.5%) (Table 2). The average nap time according to life records was 4:20:10, while the longest was 5:05:21 for fire-fighters, and the shortest was 3:12:38 for the other teams.

## 4. Discussion

Our study provides new insight into energy expenditure of a large-scale disaster training targeting a large sample size of fire-fighters belonging to multiple occupations and job ranks using the tri-axial accelerometer measurement method. The limitation that had until now prevented field studies of rural fire-fighters was the logistical difficulty and cost [10]. Substantial research has been carried out in laboratories where urban fire conditions were simulated [23]. However, rural fire conditions cannot be easily and realistically simulated in the laboratory, as the associated variables, such as the size of the area, terrain and meteorological conditions are numerous [15]. Although there are previous studies targeting forest fires, rural fires and urban fires, there are few studies targeting multiple disasters [11,12,13]. Large-scale disasters in Japan often cover wide areas, including urban areas, mountainous areas and coastal areas. The large-scale disaster response training in this study is a complex disaster activity, which includes both forest fires and urban fires. 

Fire-fighting is an intensive activity, and requires high energy (food) and physical activity. It has been reported that firefighters have felt unwell/or are having discomfort at the time of a large-scale disaster; the same has been mentioned by the fire-fighters. There is a shortage of supplies immediately after the disaster [2]. In addition, Koizumi et al. (2017) have previously reported on an insufficient preparation during a survey of fire departments nationwide, and there were voices calling for standards [4], so it was necessary to indicate the actual required amount (food in particular) by the firefighters. One approach is to estimate the energy requirements/expenditure. Although there are a few surveys on the amount of energy expenditure for regular fire-fighting work, there is no study that estimates energy expenditure during a large-scale disaster. Moreover, the energy expenditure value/data remains unclear, and thus it becomes important to first estimate the energy expenditure through a scientific study. Additionally, fire brigades are varied with different responsibilities depending on the position, which is necessary in order to clarify the command system [6]. So, we considered that the difference in energy expenditure may be related to the job type and job class. This research targets activity in large-scale disasters, and it is the primary purpose of firefighters in a large-scale disaster is to save lives, thus the TEE was determined. Moreover, as the types and tasks of rescue activities are complex and diverse, different work types were examined.

The average of TEE is about 3619 (±499) kcal. Previously, the average TEE values were reported 3626 kcal/day for a fire-fighter [11], 4878 kcal/day for a wildland fire-fighter [12], and 4009 kcal/day for an urban fire-fighter [13] using the DLW method. The AC method reported 4716 kcal/day for a wildland fire-fighter [10] and 2531 kcal/day for an urban fire-fighter [13]. In addition, as another activity field, average TEE values of professional athletes during the soccer game season were reported to be about 3532 ± 408 kcal/day [24]. In order to investigate a large number of subjects at the same time in a training that seems to be close to a real disaster, we conducted a survey using a terminal whose accuracy was confirmed in a survey comparing the accuracy of wearable devices with the chamber and DLW method [25]. However, the following underestimation is also possible for AC method. In the Touno et al. [13] study, there is also a report that there is a 33% difference between the AC method and the DLW method.

The rescue team and the other teams were found to be significantly higher with regards to the TEE than that of the firefighting team and the EMS team. The rescue team is specialized in life saving activities, and especially in the case of a large-scale disaster, it is necessary to find and rescue a person in need of removal various obstacles. Therefore, it was clear that the amount of energy expenditure had increased as the contents of activities are diverse. In addition, it may be that the energy expenditure increased because the other teams had been working for a long time, while accepting the fire-fighters without rest at the base. The others team has the shortest nap time. Robertson et al. (2017) [16] surveyed TEE for the crews performing base operations and reported a TEE of 2842 ± 649.9 kcal, which is lower than the fire-fighters 4538 ± 106.3 kcal. In the survey by Robertson et al. [16], base operations are considered to be activities that are performed when Fire Rangers are not working on fire response activities, but are reporting daily to the Fire Management Headquarters (FMH), so there may be differences with activities in this study. Touno et al. reported that the TEE of the rescue team during normal work was estimated to be 2531 ± 186 kcal/d by AC [13]. Fire fighters in the current research are considered to consume more energy than rescue teams during normal work; i.e., higher than the results obtained by Touno et al. [13]. Fire-fighting teams and EMS teams are considered to consume less energy than rescue teams and other teams. This is because the frequency of use of equipment and vehicles is high, and it is considered that standby time and nap time are longer.

Although there are previous researches that have surveyed and reported by the roles of the fire brigade personnel [16], there are few studies that have reported by job rank. Research into physiological responses during fire-fighting activities is reliant on data collected during simulations of live fires performed in facilities that are used to train newly recruited firefighters [26]. Fire-fighting personnel engage in dangerous operations such as fire suppression and life-saving at sites such as fires. In order to conduct efficient firefighting activities and ensure the safety of the staff, a system with a solid command and control system is required. In order to establish the command and command system of such fire brigade staff, unit formation under the class system is performed [27]. Therefore, it is possible that the content of the activity differs depending on the rank, and the energy expenditure also changes accordingly. In this research, since the fire Captain is responsible for instructing members, it can be assumed that they experience lower energy expenditure than the fire Lieutenant and the fire Sergeant. Among the members, the energy expenditure of the fire Captain and the fire Lieutenant was high because they thought that they carried out the activities themselves while giving their command to the subordinates, the activity time was long, and the activity contents were diverse.

Cuddy reported that Wildland fire-fighters spent 49 ± 8%, 39 ± 6%, and 12 ± 2% in the SED, LPA and MPA/VPA categories, respectively [28]. In this study, EMS teams spent 48.7%, 40.4% and 10.9% in the SED, LPA and MPA/VPA categories, similar to Cuddy’s report. The SED and LPA of EMS teams were found to be significantly larger than the rescue team, firefighting team and the other teams. In addition, EMS teams were on standby while the rescue team and firefighting team searched and transported the rescuers. The other teams have the shortest nap time and the lowest SED rate. The other teams do not have direct disaster activities, but support activities of the rescue team, firefighting team, and EMS team at any time, resulting in a low SED rate. The rescue team carried the equipment, climbing and searching for rescuers, excavating the shovel, and transporting the sand and equipment. Because there was a significant difference in the types of activities, it was considered necessary to examine the contents of each activity group when considering active foods and supplementary foods in the future.

This research also highlighted some limitations, and these are mentioned below: (a)Load by equipment: At the time of disaster activity a variety of equipment is necessary, so the load by the carrying items is added. For rescue and fire-fighters, a load of 25 kg was calculated with all attached, including respiratory equipment and EMS was estimated to have a load of 4 kg. It has been reported that energy expenditure and exercise intensity increase with load [29]. Energy expenditure is expected to increase due to the load of the items to be carried, but the increase in energy expenditure of this load cannot be measured by the AC method.(b)Fire-fighting specific actions: Accelerometers generate their output in form of “counts” per unit time [30]. In order for the accelerometer to recognize the activity, it is necessary to count, but the disaster operation includes many activities that are not counted such as maintaining the load state, activities bending the body in a collapsed building, etc. In the future, it will be necessary to add a method for measuring such activities.(c)Inactivity for training: It is probable that the activities were limited because this study was training. Disaster assumptions end after a certain amount of time, because the activity of trainings is planned and limited. In an actual disaster, the priority is saving human life, so the amount of activity may increase compared to training. The influence of the weather is one of them. It rained during training A, B and C. In training A, a part of training was interrupted due to rain, but in training B and C activity continued without interruption. Since it was equivalent weather conditions, it is expected that continuation could be possible even in training A. Particularly in emergency situations in the case of large-scale disasters, this rainfall amount (0.5 mL/h) is likely to continue activity. However, if the risk of secondary disasters is considered due to weather conditions such as sediment-related disasters caused by heavy rain, there is also the possibility of deciding to reduce the activity scale. In disaster activities, while paying attention to the safety of the crew members, they are engaged in activities with the highest priority to human life, so if the risk of secondary disasters and the possibility of danger to the crew members are low, the activities will continue even if it is raining. From the above, it seems that the interruption of rain activity in this research is an interruption because it was training. This leads to an underestimation of energy expenditure in actual disasters. In this view, there was no energy expenditure by physical activity that activity originally should take place from 15:30 to 17:00, where activity was interrupted by rain in training A. In training A, activity was interrupted for 1 h and 30 min. In addition, since rest and nap were reliably given because it was training, it is considered that energy expenditure was lower because the activity time was limited than the actual disaster activity. From the above, the results of this study, which can be measured with a 3-axis accelerometer, seem to show the lowest energy expenditure during disaster activities.(d)The number of participants and diversity of situations: The number of participants and diversity of situations were limited. It is possible that there is also a bias in the season in which the target training was conducted.

## 5. Conclusions

As a result of investigating the 5 trainings and 240 subjects using the AC method that can calculate energy expenditure of fire-fighters (disaster response team) in large-scale disaster activity training, an average TEE of 3619 (±499) kcal/24 h was determined. As an activity group, rescue teams and other teams have high energy expenditure. In the job rank, the Fire Lieutenant has high energy expenditure. Due to load, fire-fighting specific actions and activity limitations for training, the underestimation by the AC method could be considered in the future.

## Figures and Tables

**Figure 1 nutrients-13-02789-f001:**
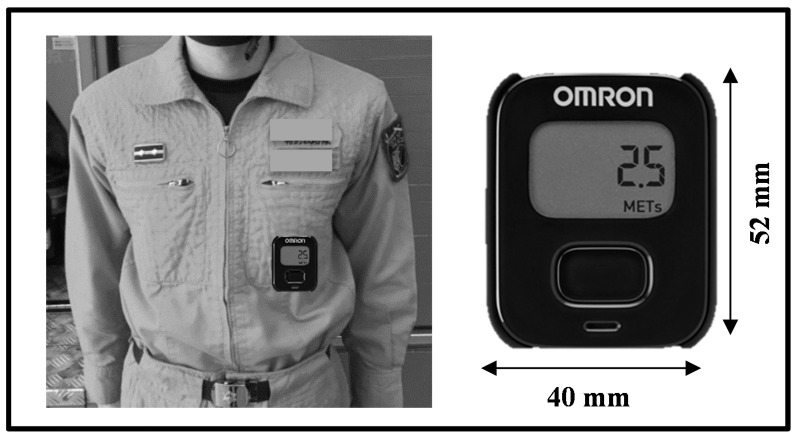
Placement of to wear a tri-axial accelerometer (Omron activity meter, Active style PRO HJA-750C—23 g, 40 × 52 × 12 mm; Omron Healthcare Co., Ltd. Kyoto, Japan). The base part of the chest pocket (length-16.2 cm, width-15.3 cm) is located near the waist area (about 10 cm above the belt part).

**Figure 2 nutrients-13-02789-f002:**
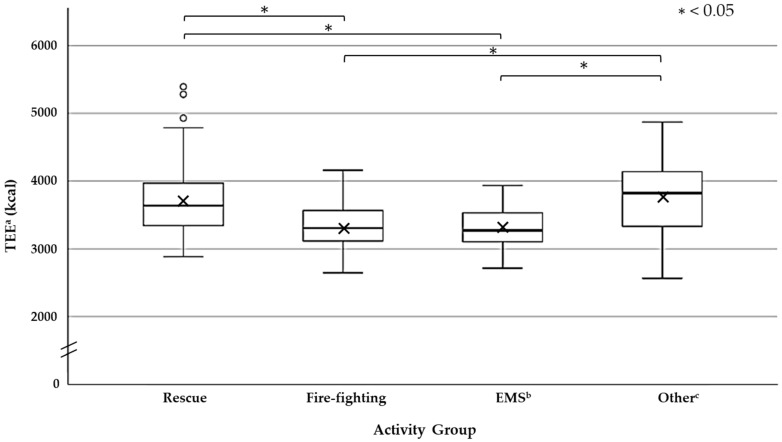
Total energy expenditure (TEE) of each disaster response Activity Group (24 h from arrival at the disaster site to the end of the disaster response activity during the two days of activity training). The TEE of the Rescue team and the other team * indicates significant (One-way analysis of variance and Tukey’s HSD method: *p* < 0.05) higher than the TEE of the Fire-Fighting team and the EMS team. × shows the average of each activity group in the box plot. ∘ shows the outlier. ^a^, TEE = Total energy expenditure, ^b^, EMS = Emergency medical service, and ^c^, Other = Command and back up unit.

**Figure 3 nutrients-13-02789-f003:**
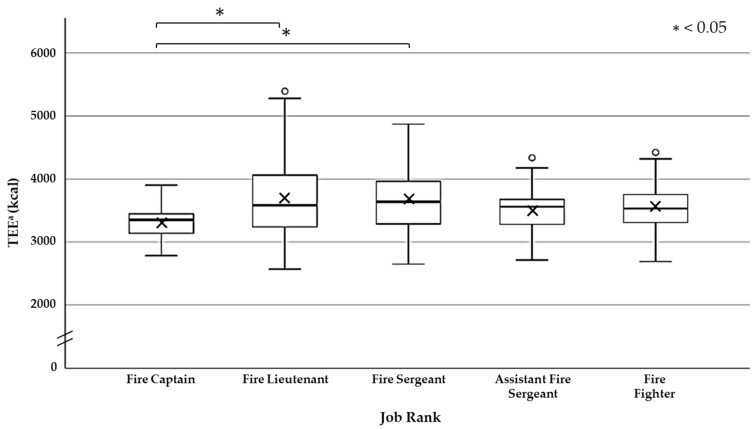
Total energy expenditure (TEE) of each Job Rank (24 h from arrival at the disaster site to the end of the disaster response activity during the two days of activity training). The TEE of the Fire Captain * indicates significant (One-way analysis of variance and Tukey’s HSD method: *p* < 0.05) lower than the TEE of the Fire Lieutenant and the Fire Sergeant. × shows the average of each activity group in the box plot. ∘ shows the outlier. ^a^ TEE = Total energy expenditure.

**Table 1 nutrients-13-02789-t001:** Activity group and job rank of participants by each disaster simulation training.

		Rescue	Fire-Fighting	EMS *	Other **	Subtotal
Training A (Oct. 2016)	Fire Captain	6	0	0	1	7
	Fire Lieutenant	11	3	4	2	20
	Fire Sergeant	11	3	2	1	17
	Assistant Fire Sergeant	5	3	1	0	9
	Firefighter	6	1	2	0	9
Subtotal		39	10	9	4	62
Training B (Mar. 2017)	Fire Captain	2	-	0	2	4
	Fire Lieutenant	17	-	1	5	23
	Fire Sergeant	3	-	0	0	3
	Assistant Fire Sergeant	0	-	0	0	0
	Firefighter	0	-	0	0	0
Subtotal		22	-	1	7	30
Training C (Oct. 2017)	Fire Captain	6	2 (1) ^†^	0	1	8
	Fire Lieutenant	10	2	2	1	15
	Fire Sergeant	16	3	3	1	23
	Assistant Fire Sergeant	1	0	2	0	3
	Firefighter	7	3	2	0	12
Subtotal		40	9	9	3	61
Training D (Mar. 2018)	Fire Captain	0	0	0	1	1
	Fire Lieutenant	17	0	0	6	23
	Fire Sergeant	4	0	0	0	4
	Assistant Fire Sergeant	0	0	0	0	0
	Firefighter	0	0	0	0	0
Subtotal		21	0	0	7	28
Training E (Nov. 2018)	Fire Captain	3	0	0	1	4
	Fire Lieutenant	10	3	3	2	18
	Fire Sergeant	12	3	2	2	19
	Assistant Fire Sergeant	6	1	4	0	11
	Firefighter	5	2	0	0	7
Subtotal		36	9	9	5	59
	Subtotal ***	158 (66%)	28 (12%)	28 (12%)	26 (11%)	
	Total	240				

Activity groups are organized rescue, firefighting, Emergency medical service (EMS) and the other teams (the command and back up unit). Each officer is classified by job rank (Fire captain, Fire Lieutenant, Fire Sergeant, Assistant Fire Sergeant, and Fire Fighter). * EMS = Emergency medical service ** Other = Command and back up unit *** Subtotal = Each activity group. ^†^ One (1) participant withdrew from the study due to mechanical trouble.

**Table 2 nutrients-13-02789-t002:** Intensity of activity Group and Average nap time.

		ALL		Rescue		Fire-Fighting		EMS *		Other **	
SED		39.0%		37.8%	C	40.5%	CD	48.7%	ABD	34.7%	BC
LPA		44.5%		44.5%	C	46.9%	C	40.4%	ABD	46.6%	C
MPA+VPA		16.5%		17.8%	BC	12.6%	AD	10.9%	AD	18.7%	BC
Average nap time (SD)	MIN ***	4:20:10 (±1:35:11)	1:00:00	4:21:36 (±1:38:51)	1:30:00	5:05:21 (±1:33:07)	3:30:00	4:12:52 (±1:00:12)	2:30:10	3:12:38 (±1:10:12)	1:00:00
	MAX ***		7:30:00		7:30:00		7:30:00		5:30:00		5:00:00

A: significant (*p* < 0.05) for Rescue; B: significant (*p* < 0.05) for Firefighting; C: significant (*p* < 0.05) for EMS; D: significant (*p* < 0.05) for Other. The intensity of activity Group was classified into SED, LPA and MPA+VPA by each activity group (activity count cut points for sedentary (SED; >1.49 METs), light (LPA; >1.5–2.99 METs), moderate (MPA; >3.0–5.99 METs), and vigorous-intensity physical activity (VPA; >6 METs)). A, B, C and D * indicate significant (One-way analysis of variance and Tukey’s HSD method: *p* < 0.05). Average nap time is according to the life records. * EMS = Emergency medical service ** Other = Command and back up unit. *** MIN and MAX are expressed as (hh: mm: ss).

## Data Availability

The data presented in this study are available on request from the corresponding author.

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
