# Peer review of "Estimating Total Energy Expenditure for Fire-Fighters during Large Scale Disaster Response Training Using a Tri-Axial Accelerometer"

_nutrients, 2021, doi:10.3390/nu13082789_

Round 1

Reviewer 1 Report

This study presents energy expenditure of personnel working in rescue missions during natural disasters. It’s an interesting and novel set of data, collected during a rare opportunity. However, the significance and implications of such data need to be drawn out further in the manuscript.

Abstract:

  • Who is the “Activity group” referring to?
  • Spell out EMS at first appearance.

Introduction:

  • The introduction gives sufficient context and background relating to frequency of natural disasters in Japan, and the role of rescue teams during natural disasters.
  • However, this section can be improved by highlighting why it is so important to determine the TEE of fire fighters during rescue missions, and why it is necessary to determine the difference between fire fighters of different job ranks. The studies quoted from line 45 – 49 is not quite adequate to support the significance of this study’s research question.
  • Avoid using “blinded” when describing the study (line 113) and in the abstract, as this may confuse the reader, that the study is a blinded intervention.

Methods:

  • Participant characteristics under section 2.2.2 (including Table 1) should be moved to the Results section.
  • Were there any eligibility criteria for participant selection? This needs to be outlined.
  • How was the study described to the participants? Was it compulsory to wear the AC as part of the training?
  • How long did the fire fighters wear the AC for, during each blind training?
  • BMR was written as BRM in line 198.
  • How was DIT calculated?
  • It is unclear why the values had to be converted to AEE (then take away BMR), when in the next step, it’s added together again to calculate TEE.

Results

  • Table 1: this data will be better presented as frequency (%) rather than a tally.
  • Consider separating the commander and back-up units in the analysis. As results show the “other” group had the highest TEE, and I suspect the commander and back-up units will have different activity levels during the rescue mission.
  • Nap time data presented in Table 2 was difficult to understand

Discussion

  • Line 277 – 288: What was the purpose of the comparisons with the other studies? To prove that this study’s results are reliable or unreliable?
  • Lin 315 – 322: “Energy consumption” should not be used interchangeably with “energy expenditure”. “Energy consumption” means energy intake through eating, which was not measured in this study.
  • The significance of this study’s results needs to be pinpointed and elaborated. Why is it important to determine the TEE of rescue teams? Is it to ensure sufficient food supply is provided? Are they currently not provided – hence a gap in the literature? The implications of this study’s results should form a key part of the discussion.

Reviewer 2 Report

Please closely follow the STROBE guidelines for reporting observational studies.

Approach:

This is an observational study. Was it cross-sectional? Repeated measures?

Participants:

Eligibility criteria are not listed.

Why were there only males included?

Recruitment: How? When? By whom?

What was the size of the target population from which the sample was selected?

What does “blind training” mean?

Procedures:

Why was body mass and body weight obtained via self-report? Discuss the validity of this approach and provide a citation.

Activity log:

More details are needed. For example - Is this a reliable and validate measure? Provide a citation.

Outcome Measures:

What was the primary outcome?

How was sample size determined? Describe power analysis to determine sample size.

Discussion:

What is the key finding of this study?

Discuss limitations of this study.

Discuss practical applications.

Discuss future research.

Reviewer 3 Report

I would like to thank the editorial board for the opportunity to review the current manuscript that was titled “Estimating Total Energy Expenditure for Fire-Fighters during Large Scale Disaster Response Training using a Tri-axial Accelerometer”. The current paper makes a valuable contribution to understand the energy expenditure of different ranks during potential disaster responses. This manuscript will be very useful for identifying the food requiresment during these disaster situations.

General Comment

The manuscript is well presented with informative figures and tables to present the data. There are a number of minor edits to be addressed. I have included some comments below. I also suggest another proof of the manuscript as there are a number of grammar errors. I have attempted to identify most of them but there are some others that may be observed when a final proof of the document is undertaken.

Abstract

Line 20 – “athletes during game season” this statement is vague, what sport are you referring to? For example, update to include the sport, .......athletes during a (insert sport....soccer) game.

Line 21 – identify what “EMS” is.

Line 22 – please include the difference between the Fire Captain, Fire Lieutenant and Fire Sergeant groups, for example include the effect size

Introduction

Overall the introduction is clear for the reader to understand the background for the study. Some minor edits are required:

L33 – delete “the”.... critical because the priority is on saving the victims' lives....

L45 – seems to be a word missing..... this lack of creates an adverse effect..... this lack of food or energy creates an adverse effect....

L48 – delete..... “the”... about half of fire-fighting personnel....

L51 – update..... to make available food, water, personal equipment, etc., for more than 72 hours..... to make food, water, personal equipment, etc., available for more than 72 hours

...

Methods

L138 – ...... They organize rescue, firefighting, EMS and the other teams (the command and back up 138 unit).... this sentence is not clear? Are you stating how the groups were divided? Please update

L164 – update.... so the equipment is possibility of breakage when equipment attached to the outside and possibility of detachment unintentionally......... so there is a  possibility of breaking the equipment when it is attached outside and a possibility of it detaching unintentionally

L186 – it appears that you intended to include the name of the authors in the sentence but the reference is included instead?? ....... have been widely used [17] have been recommended.... please update

Results

Table 2 – I understand that the table may have been adjusted when you placed into the document template or onto the online system. Please update the Nap section as it is difficult to read in the current layout

Discussion

L 268 – please mention that it is the current results or the current study..... For the energy intake, it was reported that the fire-fighters.... The current results showed that fire fighters consumed an inadequate amount of total calories....

L281-282 – please identify the sport of the professional athletes

L 294 – you mention that it was “clear”. This is a strong assumption. I suggest that you replace “clear” with “may” ........ In addition, the energy consumption may have increased because the other teams

L 298 – the fire fighters results, are you referring to the current study? If so please mention this

L 316-318 – this sentence is not clear as it is written.  In this research, since the fire Captain is responsible for instructing and instructing members, it can be considered that it showed lower energy consumption than the fire Lieutenant and the fire Sergeant.... maybe replace with..... In this research, since the fire Captain is responsible for instructing members, it can be assumed that they experience a lower energy consumption than the fire Lieutenant and the fire Sergeant

L 318-321 – Are you referring to the current study? If so please mention that

L325 – please include “respectively”.... 10.9% in the SED, LPA, and MPA/VPA categories, respectively.....

L326 – what was “significantly shorter” ? please identify what unit of measurement you are referring to

L 352 – remove “to” ...... priority is saving human life...

Round 2

Reviewer 2 Report

The authors made substantial upgrades to the manuscript. However, the manuscript does not follow the STROBE guidleiens for reporting observational studies. The STROBE statement and checklists are found here:

https://www.strobe-statement.org/index.php?id=available-checklists
